# Dose escalation and expansion cohorts in patients with advanced breast cancer in a Phase I study of the CDK7-inhibitor samuraciclib

Samuraciclib is a selective oral CDK7-inhibitor. A multi-modular, open-label Phase I study to evaluate safety and tolerability of samuraciclib in patients with advanced malignancies was designed (ClinicalTrials.gov: NCT03363893). Here we report results from dose escalation and 2 expansion cohorts: Module 1A dose escalation with paired biopsy cohort in advanced solid tumor patients, Module 1B-1 triple negative breast cancer (TNBC) monotherapy expansion, and Module 2A fulvestrant combination in HR+/HER2− breast cancer patients post-CDK4/6-inhibitor. Core study primary endpoints are safety and tolerability, and secondary endpoints are pharmacokinetics (PK), pharmacodynamic (PD) activity, and anti-tumor activity. Common adverse events are low grade nausea, vomiting, and diarrhea. Maximum tolerated dose is 360 mg once daily. PK demonstrates dose proportionality (120 mg-480 mg), a half-life of approximately 75 hours, and no fulvestrant interaction. In dose escalation, one partial response (PR) is identified with disease control rate of 53% (19/36) and reduction of phosphorylated RNA polymerase II, a substrate of CDK7, in circulating lymphocytes and tumor tissue. In TNBC expansion, one PR (duration 337 days) and clinical benefit rate at 24 weeks (CBR) of 20.0% (4/20) is achieved. In combination with fulvestrant, 3 patients achieve PR with CBR 36.0% (9/25); in patients without detectable TP53-mutation CBR is 47.4% (9/19). In this study, samuraciclib exhibits tolerable safety and PK is supportive of once-daily oral administration. Clinical activity in TNBC and HR+/HER2- breast cancer post-CDK4/6-inhibitor settings warrants further evaluation.

Cyclin-dependent kinases (CDKs) are rational targets for cancer therapy due to their important roles in cell division and transcription[1]. CDK7 plays a key role in regulation of the cell cycle as the CDK activating kinase (CAK) responsible for phosphorylating cell cycle CDKs, which promotes association with their cognate cyclin and/or enhances kinase activity[2]. CDK7 is also required for transcriptional control by (1) initiating transcription initiation by phosphorylating RNA polymerase II and (2) regulating enhancer activities by phosphorylating many transcription factors such as nuclear hormone receptor, including estrogen and androgen receptors, leading to their activation[2–4], in breast and prostate cancer and is implicated in resistance to endocrine therapies. CDK7 is over-expressed in several cancers and its expression is associated with poor prognosis[5].

Pre-clinical studies have demonstrated the sensitivity of many cancers to selective inhibitors of CDK7[6,7]. Samuraciclib (ICEC0942; CT7001) is a potent small molecule, adenosine triphosphate (ATP)

✉ e-mail: matthew.krebs@manchester.ac.uk

competitive inhibitor of CDK7[8] (Supplementary Fig. 2). Pre-clinical studies have shown that CDK7 inhibitors, including samuraciclib, are effective in both hormone receptor positive (HR+) and triple negative breast cancer (TNBC). In HR+ breast cancer, samuraciclib is effective both alone, and when combined with hormonal therapy, in breast cancer models, and that CDK7 inhibition would be effective even after resistance develops to CDK4/6 inhibitors[9]. Additionally, in HR+ breast cancer, preclinical data indicate that CDK7 inhibition activates the p53 pathway in TP53 WT cancer cell lines, inducing apoptosis[10,11]. In TNBC, several studies have confirmed the initial observation[10] of sensitivity to CDK7 inhibition;[5,12,13] with encouraging activity with samuraciclib observed in vivo, including in patient-derived xenograft models of TNBC[14]. Preclinical pharmacokinetic (PK) studies with samuraciclib showed good oral bioavailability in mice, rats and dogs.

This modular Phase I study was designed to establish the optimal dose of samuraciclib when used as monotherapy or in combination with other anti-cancer treatments. The results from dose escalation and 2 expansion cohorts in breast cancer are presented here. Module 1A was a first-in-human dose escalation study to assess initial safety, tolerability and the PK profile of samuraciclib and to identify the maximum tolerated dose (MTD). Module 1 Part B, Part 1 in TNBC (Module 1B-1), was designed to refine the safety, tolerability, PK, and pharmacodynamic profiles of samuraciclib monotherapy (360 mg once daily [OD]) in patients who had received prior systemic therapy for advanced TNBC. Module 2A explored the safety, tolerability and preliminary efficacy of 2 dose levels of samuraciclib (240 mg OD and 360 mg OD) in combination with fulvestrant in HR+/HER2− advanced breast cancer patients who had previously received CDK4/6 inhibitor therapy.

In this work, samuraciclib has an acceptable safety profile and initial evidence of efficacy is demonstrated as a selective inhibitor of CDK7. Samuraciclib has the potential to address the significant medical need of patients whose disease has progressed on CDK4/6 inhibitors and will be investigated further in future studies.

## Results

### Patient disposition
In Module 1A, 33 patients with advanced/metastatic solid tumors were enrolled. Patients initially received samuraciclib at a starting dose of 120 mg, with 4 additional dose levels of 240 mg OD, 360 mg OD, 480 mg OD and 180 mg twice daily (BID) explored. Eleven patients were recruited to a breast cancer expansion cohort to evaluate the pharmacodynamic effects of samuraciclib in tumor tissue.

In Module 1B-1, 23 patients with locally advanced and/or metastatic TNBC received samuraciclib 360 mg OD.

In Module 2A, 31 patients with post-CDK4/6 inhibitor HR+/HER2− advanced breast cancer received samuraciclib in combination with fulvestrant: 6 at 240 mg OD and 25 at 360 mg OD of samuraciclib (Supplementary Fig. 1).

### Demography and baseline characteristics
In Module 1A, mean age was 59.6 years and 63.6% of patients were female (Table 1). Primary malignancies were predominantly breast (30.3%) or colorectal (24.2%). 87.9% of patients had received prior chemotherapy and 36.4% had undergone hormone therapy. In the paired biopsy cohort, mean age was 56.5 years, all patients had undergone prior chemotherapy and 90.9% had received prior hormone therapy.

In Module 1B-1, the mean age was 53.6 years and all patients were female. Patients had received a median of 2 (range 1-3) lines of prior chemotherapy in the advanced TNBC setting.

In Module 2A, the mean age was 60.4 years, and all patients were female. All patients had received prior aromatase inhibitor (AI) in combination with CDK4/6. Six of 31 patients were pre-menopausal (all were receiving goserelin).

## Table 1 | Demographic details

| | Module 1A—dose escalation N = 33 | Module 1A—paired biopsy N = 11 | Module 1B-1 N = 23 | Module 2A N = 31 |
|---|---|---|---|---|
| **Demographic details** | | | | |
| Age, years (range) | 59.6 (19–78) | 56.5 (26–75) | 53.6 (32–75) | 60.4 (41–81) |
| Sex, male/female (n, %) | 12 (36.4)/ 21 (63.6) | 0/11 (100.0) | 0/23 (100.0) | 0/31 (100.0) |
| **Race, n (%)** | | | | |
| White | 28 (84.8) | 11 (100.0) | 19 (82.6) | 26 (83.9) |
| Black | 0 | 0 | 0 | 2 (6.5) |
| Asian | 3 (9.1) | 0 | 0 | 2 (6.5) |
| Other | 2 (6.1) | 0 | 4 (17.4) | 1 (3.2) |
| **Cancer type** | | | | |
| Breast | 10 (30.3) | 11 (100.0) | 23 (100) | 31 (100.0) |
| Colorectal | 8 (24.2) | 0 | 0 | 0 |
| Liver | 1 (3.0) | 0 | 0 | 0 |
| Lung | 2 (6.1) | 0 | 0 | 0 |
| Pancreas | 1 (3.0) | 0 | 0 | 0 |
| Prostate | 2 (6.1) | 0 | 0 | 0 |
| Stomach | 1 (3.0) | 0 | 0 | 0 |
| Other | 8 (24.2) | 0 | 0 | 0 |
| **Metastases, n (%)** | 27 (81.8) | 9 (81.8) | 21 (91.3) | 31 (100.0) |
| **Prior therapies, n (%)** | | | | |
| Chemotherapy | 29 (87.9) | 11 (100.0) | 23 (100.0) | 16 (51.6%) |
| Radiotherapy | 10 (30.3) | 8 (72.7) | 21 (91.3) | 21 (67.7) |
| Immunotherapy | 1 (3.0) | 0 | N/A | 3 (9.7) |
| Hormone therapy | 12 (36.4) | 10 (90.9) | 7 (30.4) | 31 (100.0) |
| CDK4/6 inhibitor therapy | 1 (3.0) | 0 | N/A | 31 (100.0) |
| Surgery | 3 (9.1) | 5 (45.5) | 23 (100.0) | 25 (80.6) |
| Other | 19 (57.6) | 8 (72.7) | N/A | 0 |
| Biological/immunological/other | N/A | N/A | 8 (34.8) | N/A |

N/A not applicable.

### Evaluation of dose levels
No dose-limiting toxicity (DLTs) were observed at 120 mg and 240 mg OD monotherapy doses, so the dose was escalated to 480 mg OD. Four patients had gastrointestinal DLTs at this level, so this dose was considered not tolerated (Supplementary Table 5) and dose was reduced to 360 mg OD. No DLTs were observed at this level but approximately 50% of patients experienced some gastrointestinal symptoms so a split dosing regimen of 180 mg BID was explored. In this group DLTs were observed in 2 patients (gastrointestinal events in 1 patient and hematological toxicity in the other). 360 mg OD dose was therefore determined to be the MTD.

### Safety results
Common AEs (frequency ≥10%) and AEs ≥Grade 3, regardless of relationship to study treatment, are shown in Table 2. 96/98 patients (98.0%) had at least one AE considered related to samuraciclib (Supplementary Tables 8, 9, and 10), with these patients having at least one drug-related gastrointestinal AE, primarily diarrhea, nausea and vomiting. No difference in the AE profile was seen between the 240 mg dose and the 360 mg dose in combination with fulvestrant, and no obvious dose-related trends were seen (Supplementary Table 10). The majority of events were low grade, reversible, and manageable using standard medication or dose reductions.

**Table 2 | Treatment-emergent adverse events (reported in ≥10% of patients in each study)**

| MedDRA preferred term Number of patients (%) | Module 1A N = 44 | | Module 1B-1 N = 23 | | Module 2A N = 31 | |
|---|---|---|---|---|---|---|
| | All AEs | Grade ≥3 | All AEs | Grade ≥3 | All AEs | Grade ≥3 |
| Any treatment-emergent AE | 44 (100.0) | 21 (47.7) | 23 (100.0) | 10 (43.5) | 31 (100.0) | 21 (67.7) |
| Diarrhea | 38 (86.4) | 2 (4.5) | 21 (91.3) | 3 (13.0) | 28 (90.3) | 6 (19.4) |
| Vomiting | 36 (81.8) | 1 (2.3) | 14 (60.9) | 2 (8.7) | 26 (83.9) | 1 (3.2) |
| Nausea | 34 (77.3) | 0 | 22 (95.7) | 1 (4.3) | 26 (83.9) | 3 (9.7) |
| Fatigue | 17 (38.6) | 0 | 11 (47.8) | 1 (4.3) | 15 (48.4) | 1 (3.2) |
| Abdominal pain | 12 (27.3) | 0 | 6 (26.1) | 0 | 8 (25.8) | 0 |
| Anemia | 10 (22.7) | 2 (4.5) | 0 | 1 (4.3) | 4 (12.9) | 3 (9.7) |
| Decreased appetite | 9 (20.5) | 1 (2.3) | 3 (13.0) | 0 | 13 (41.9) | 0 |
| ALT increased | 8 (18.2) | 1 (2.3) | 0 | 0 | 7 (22.6) | 2 (6.5) |
| Cough | 8 (18.2) | 0 | 3 (13.0) | 0 | 3 (9.7) | 1 (3.2) |
| Upper respiratory tract infection | 8 (18.2) | 0 | 3 (13.0) | 0 | 0 | 0 |
| Constipation | 7 (15.9) | 0 | 7 (30.4) | 0 | 8 (25.8) | 0 |
| AST increased | 7 (15.9) | 0 | 0 | 0 | 7 (22.6) | 1 (3.2) |
| Dyspnea | 6 (13.6) | 2 (4.5) | 0 | 0 | 5 (16.1) | 2 (6.5) |
| Back pain | 6 (13.6) | 0 | 0 | 0 | 2 (6.5) | 0 |
| Urinary tract infection | 5 (11.4) | 0 | 0 | 0 | 3 (9.7) | 0 |
| Abdominal pain upper | 4 (9.1) | 0 | 0 | 0 | 8 (25.8) | 0 |
| Headache | 4 (9.1) | 0 | 0 | 0 | 8 (25.8) | 0 |
| Weight decreased | 4 (9.1) | 1 (2.3) | 0 | 0 | 4 (12.9) | 0 |
| Dizziness | 2 (4.5) | 0 | 0 | 0 | 5 (16.1) | 0 |
| Dysphagia | 2 (4.5) | 0 | 0 | 0 | 4 (12.9) | 0 |
| Hypocalcaemia | 2 (4.5) | 0 | 0 | 0 | 4 (12.9) | 0 |
| Rash | 1 (2.3) | 0 | 0 | 0 | 6 (19.4) | 1 (3.2) |
| Hyperglycemia | 1 (2.3) | 0 | 0 | 0 | 5 (16.1) | 0 |
| Stomatitis | 1 (2.3) | 0 | 3 (13.0) | 1 (4.3) | 4 (12.9) | 0 |
| Dysgeusia | 1 (2.3) | 0 | 0 | 0 | 4 (12.9) | 0 |
| Hypokalaemia | 0 | 0 | 0 | 0 | 5 (16.1) | 2 (6.5) |
| Taste disorder | 0 | 0 | 0 | 0 | 4 (12.9) | 0 |

Patients with multiple incidences of the same AE are counted once for each preferred term. Includes AEs with an onset date on or after the date of first dose and up to and including 28 days following the date of last dose of study medication.

*AE* adverse event, *ALT* alanine aminotransferase, *AST* aspartate aminotransferase, *MedDRA* Medical Dictionary for Regulatory Activities.

In the dose escalation phase of Module 1A, the severity of treatment-emergent adverse events (TEAEs) increased with dose (Supplementary Tables 6 and 7). In Module 1A, discontinuations were most frequent in the highest (non-tolerated) dose cohort of 480 mg OD. At the monotherapy doses considered clinically active (240 mg OD and 360 mg OD), only 4% (1/24) discontinued treatment due to a related AE (nausea). In Module 1B-1, 1/23 (4.3%) discontinued samuraciclib due to 2 related AEs of diarrhea and recurrent nausea (both Grade 1). In combination with fulvestrant, 7/31 patients (23%) across both dose levels discontinued due to related gastrointestinal AEs (all Grade 2 or 3) and 5 discontinued due to unrelated AEs (1 each of COVID

infection [Grade 4], brain metastases [Grade 3], diaphragmatic hernia [Grade 3], anxiety [Grade 3] and hypercalcemia from bone metastases [Grade 2]). The discontinuations were similar in the 2 dose groups.

Across both parts of Module 1 and Module 2A, 5 patients died following an AE (2 patients in Module 1A - pulmonary metastases and diaphragm muscle weakness [both considered disease progression], 2 patients in Module 1B-1 – 1 dyspnea and metastases to central nervous system, [both considered due to disease progression], and 1 patient in Module 2A due to cerebral hemorrhage [in the absence of thrombocytopenia]). None of the deaths were considered related to samuraciclib treatment. Serious adverse events (SAEs) (not including deaths) were reported in 27 patients, with 6 considered related to study treatment (2 diarrhea [1 at 180 mg BID dose and 1 at 240 mg OD dose], 1 thrombocytopenia [180 mg BID dose], 1 liver injury in a patient with documented liver metastases [240 mg OD dose], and 1 esophagitis and gastroesophageal reflux disease [180 mg BID dose] and one patient with 4 events - anemia, diarrhea, dyspnea at rest, and thrombocytopenia [360 mg OD dose]). The 180 mg BID dose regimen was ultimately declared non-tolerable and not explored further.

One observation from hematological analysis in Module 1A was a reduction in platelet counts during therapy, with an approximate 20% drop in platelet counts being seen. This appeared over the first 15 days on study and then plateaued for the duration of treatment (Supplementary Fig. 3a). In the majority of patients, platelet counts remained within the normal range, were not associated with bleeding events, and resolved when treatment ended. Thrombocytopenia was reported as Grade ≥3 in 2 patients. One patient, receiving samuraciclib 180 mg BID, experienced a Grade 4 event that required transfusion with 1 unit of platelets. The patient had a pre-existing Grade 1 thrombocytopenia prior to dosing. A second patient receiving samuraciclib 360 mg OD experienced Grade 3 thrombocytopenia. For both patients the events were resolving to Grade 1 by 11 days and 23 days, respectively, after samuraciclib dosing ended. This finding of thrombocytopenia was subsequently confirmed in Module 1B-1 and Module 2A, with evidence of recovery within 28 days of stopping samuraciclib treatment (Supplementary Fig. 3b, c).

### Anti-tumor activity

In Module 1A, a total of 36 patients were evaluable for efficacy, with a variety of pre-treated advanced solid malignancies (Table 3). The disease control rate (DCR) was 52.8%. Fourteen patients (38.9%) maintained stable disease (SD) ≥ 12 weeks. A heavily pre-treated patient with HR+ breast cancer receiving samuraciclib monotherapy 240 mg OD achieved a partial response (PR) sustained for over 24 weeks. The percentage change from baseline and the best objective response for each patient is shown in Fig. 1A and the time on study is shown in Fig. 1B. Two castrate-resistant prostate cancer (CRPC) patients whose disease progressed with surgical/medical castration and bicalutamide/abiraterone treatment had durable prostate specific antigen (PSA) reductions of 27 weeks and 45 weeks, with reductions in PSA compared to baseline of 28% and 44%, respectively.

In Module 1B-1, a total of 20 patients were evaluable for efficacy, one PR was reported with a duration of response (DoR) of 337 days (Table 3). The percentage change from baseline and the best objective response for each patient is shown in Fig. 2A and the time on study is shown in Fig. 2B. For all other patients the best overall response (BOR) was SD (11 patients, 55.0%) or progressive disease (PD) (8 patients, 40.0%), giving an objective response rate of 5.0% (95% CI [0.13, 24.87]). In total, 4 patients (20%) achieved clinical benefit with either a PR or SD for ≥24 weeks (1 patient achieving a PR [5.0%]), giving a clinical benefit rate (CBR) of 20.0% (95% CI [5.73, 43.66]). Additionally, 1 patient had Response Evaluation Criteria in Solid Tumors (RECIST) progression at 16 weeks but continued treatment beyond progression due to ongoing clinical benefit until 64 weeks until a new brain lesion led to further progression.

**Table 3 | Summary of response rates**

| Module 1A (evaluable for anti-tumor activity analysis population) N = 36 | |
|---|---|
| Disease control rate, n (%) | 19 (52.8) |
| Objective response rate, n (%) | 1 (3.2) |
| Best objective response, n (%) | |
| Complete response | 0 |
| Partial response | 1 (2.8) |
| Stable disease | 18 (50.0) |
| Progressive disease | 17 (47.2) |
| **Module 1B-1 (evaluable for response population) N = 20** | |
| Clinical benefit rate at 24 weeks, n (%) | 5 (25.0) |
| Objective response rate, n (%) | 1 (5.0) |
| Best objective response, n (%) | |
| Complete response | 0 |
| Partial response | 1 (5.0) |
| Stable disease | 11 (55.0) |
| Progressive disease | 8 (40.0) |
| Progression-free survival (months) (Intent-to-Treat Population) N = 23 | |
| Median (95% confidence interval) | 2.4 (1.9, 3.8) |
| **Module 2A (evaluable for response population) N = 25** | |
| Clinical benefit rate at 24 weeks, n (%) | 9 (36.0) |
| Objective response rate, n (%) | 3 (12.0) |
| Best objective response, n (%) | |
| Complete response | 0 |
| Partial response | 3 (12.0) |
| Stable disease | 13 (52.0) |
| Progressive disease | 9 (36.0) |
| Clinical benefit rate at 24 weeks by subgroup, n (%) | |
| No liver metastases (N = 11) | 6 (54.5) |
| Liver metastases (N = 14) | 3 (21.4) |
| No TP53 mutation (N = 19) | 9 (47.4) |
| TP53 mutation (N = 6) | 0 |
| Progression-free survival (months) (intent-to-treat population) N = 31 | |
| Median (95% confidence interval) | 3.7 (1.8, 7.4) |
| No liver metastases (N = 17) | 13.8 (7.4, NC) |
| Liver metastases (N = 14) | 2.8 (1.8, 7.4) |
| No TP53 mutation (N = 20) | 7.4 (3.7, NC) |
| TP53 mutation (N = 7) | 1.8 (1.7, NC) |

Disease control rate defined as percentage of patients with a complete response (CR) or partial response (PR) or stabilization of disease at first on-treatment RECIST assessment. Objective response rate defined as the percentage of patients who had at least 1 objective response (CR or PR) prior to any evidence of progression. Clinical benefit rate defined as the percentage of patients with CR or PR or stabilization of disease for at least 24 weeks between enrollment and disease progression or death due to any cause. RECIST V1.1 endpoints were assessed in the response evaluable population - defined as all patients who received ≥1 dose of samuraciclib and had a post-baseline tumor assessment. Statistical analyses of progression-free survival using the Kaplan–Meier method were performed on the intent-to-treat population.
NC not calculated.

In Module 2A, 25 patients were evaluable for response, with best RECIST responses of PR in 3 (12.0%) patients (2 confirmed, 1 unconfirmed) and SD in 13 (52.0%) patients (Table 3). For the overall population, CBR was 36.0% (9/25) and median progression-free survival (PFS) was 3.7 months. The percentage change from baseline and the best objective response for each patient is shown in Fig. 3A and the time on study is shown in Fig. 3B.

A protocol-specified exploratory analysis was performed of clinicopathological factors associated with sensitivity to treatment including demographic data, number of prior therapies, sites of metastatic disease, and mutational data from circulating tumor DNA

(ctDNA), specifically TP53 status (Supplementary Information). Two parameters were found to be associated with increased likelihood of benefit in univariate analysis: absence of a detected TP53 mutation and the absence of liver metastases. There were 27 patients with TP53 mutation data available from baseline ctDNA (mutation: n = 7 and no mutation: n = 20). Patients with no detected TP53 mutation had longer PFS (7.4 months) compared to patients with detected TP53 mutation (1.8 months), with a hazard ratio of 0.14 for the absence of TP53 mutation (95% CI: 0.05-0.45, p value < 0.001) (Supplementary Fig. 4a). Twenty-five patients with TP53 mutation data had response data (mutation: n = 6 and no mutation: n = 19). There were no patients with clinical benefit in the TP53 mutation group (0% CBR) and 9 patients with clinical benefit in the group with no detectable TP53 mutation (47.4% CBR). In contrast, for all other genes where ≥4 patients with ctDNA mutations were found in the cohort (PIK3CA, ESR1, MUC16, MED12L), there were no significant differences in PFS for patients carrying mutations compared to those without. For liver metastases, there were 14 patients with liver metastases at baseline and 17 patients with no detectable liver metastases at baseline. Patients with no detectable liver metastases had longer PFS (13.8 months) compared to patients with liver metastases (2.8 months), with a hazard ratio of 0.16 (95% CI: 0.04–0.59, p value < 0.003) (Supplementary Fig. 4b). Twenty-five patients had response data (liver metastases: n = 14 and no liver metastases: n = 11). The CBR was 54.5% (6/11) for patients with no liver metastases and 21.4% (3/14) for those with liver metastases.

### Pharmacodynamic analyses
In Module 1A, a flow cytometry assay across all doses showed a significant reduction of phosphorylated RNA polymerase II (pPolII) (substrate of CDK7) of approximately 30% in lymphocytes (Supplementary Fig. 5). A reduction in pPolII was also seen in tumor tissue obtained from the paired biopsy cohort (Fig. 4). Levels of pCDK1/2/3 in tumors also showed a reduction, although this did not reach statistical significance, possibly due to high variability of this marker at baseline (Supplementary Table 11). These data are in keeping with samuraciclib causing inhibition of CDK7 activity in tumors. The effect on markers was not greater at 360 mg OD compared to 240 mg OD, indicating both doses are pharmacologically active.

### Pharmacokinetic results
Results from Module 1A demonstrated a half-life for samuraciclib of approximately 75 h after single dosing, supporting OD dosing. Samuraciclib had moderate to high apparent clearance and was extensively distributed. Dose proportionality in exposure (120–480 mg) was observed after single and multiple dosing. Samuraciclib PK appeared time independent, although this was based on a small sample size. Steady state was achieved between 8 and 15 days of dosing (Supplementary Figs. 6, 7, and 8).

Results from Module 1B-1 demonstrated that steady state was achieved by Cycle 1 Day 8 and PK trough levels then remained consistent throughout the study module following multiple dosing. The geometric mean trough plasma concentration was 39.22 ng/mL.

Analysis of the trough PK samuraciclib data from Module 2A, along with Module 1A trough PK and fulvestrant trough PK data and previously reported data[15], did not indicate an interaction between samuraciclib and fulvestrant PK nor between PK and toxicity (Supplementary Figs. 9 and 10).

### Discussion
Samuraciclib is an orally bioavailable, potent, selective inhibitor of CDK7. Previous studies have shown that samuraciclib has 17-fold selectivity over the next most sensitive kinase[8].

The results from both modules demonstrated an acceptable safety profile and evidence of clinical activity for samuraciclib in a

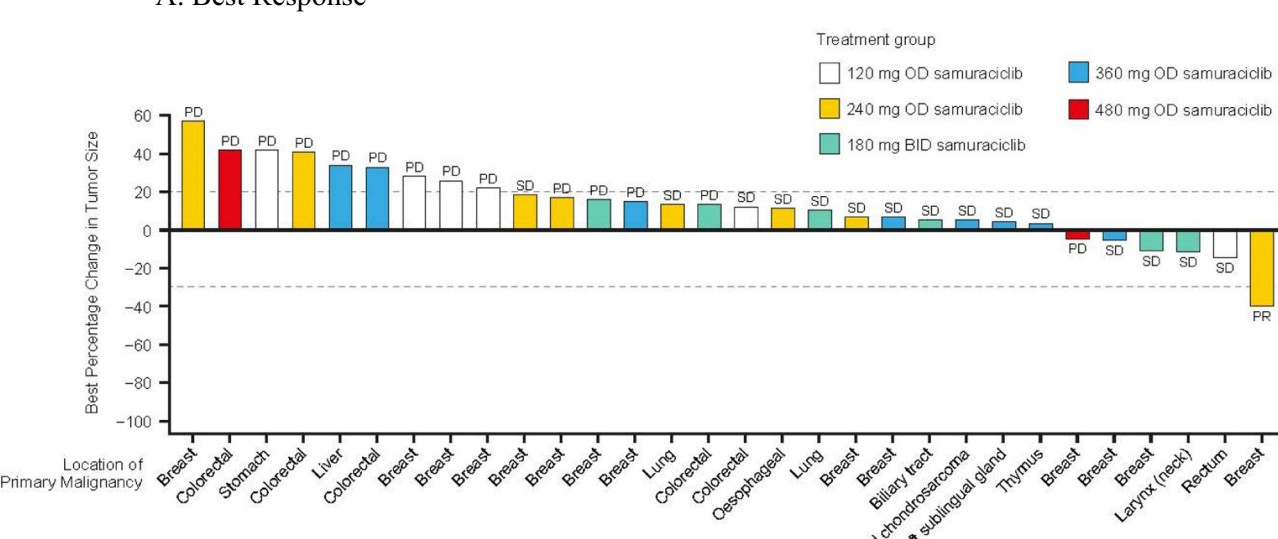

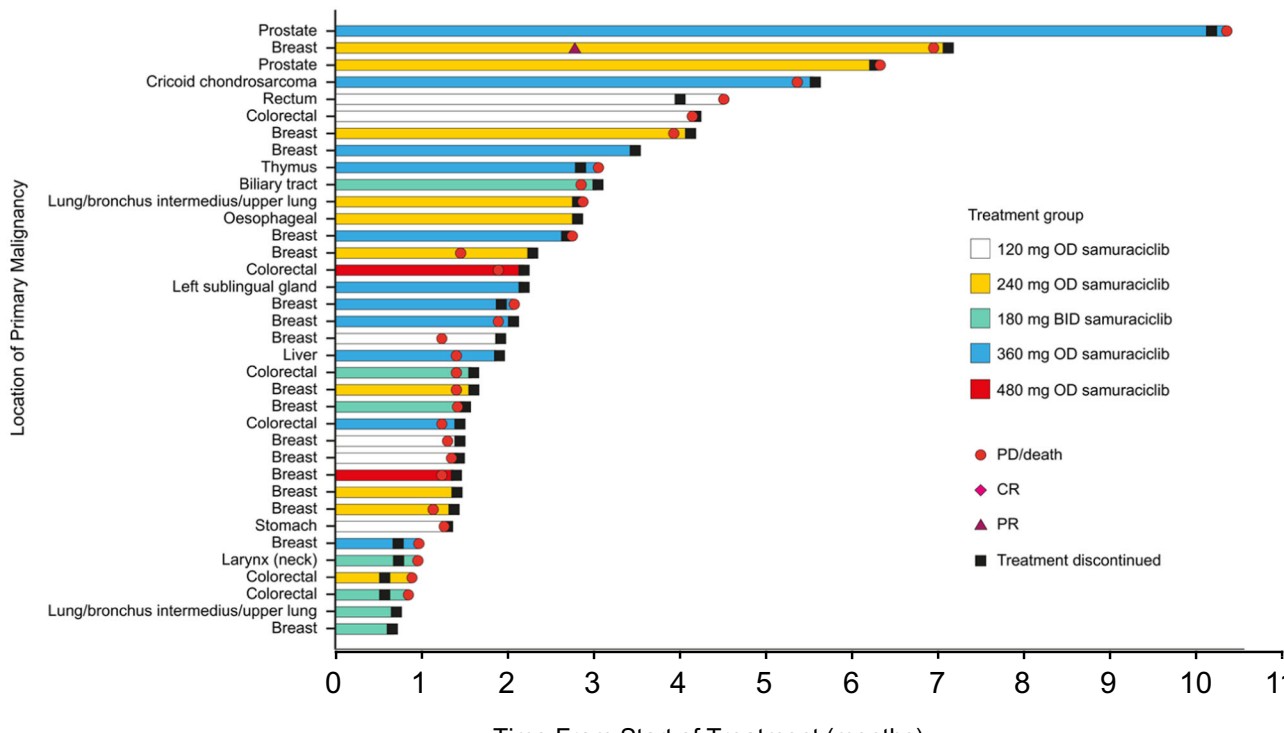

**Fig. 1 | Best RECIST response and time on study in Module 1A. A** Best response. Each bar represents the best percentage change from baseline for an individual patient and labels indicate the best objective response for the same patient. *N* = 30 (Note: only participants with measurable lesions at both baseline and post-baseline are included in this figure). BID = twice daily; OD = once daily; PD = progressive disease; PR=partial response; SD = stable disease. **B** Time on study. Each bar is annotated with the location of the primary malignancy for each participant.Only patients with measurable lesions at both baseline and post-baseline are included in the figure. *N* = 36. BID twice daily, CR complete response, OD once daily, PD progressive disease, PR partial response, SD stable disease.

variety of advanced solid malignancies, in particular patients with advanced/metastatic TNBC and HR+/HER2− breast cancer. Common drug-related AEs were gastrointestinal events (diarrhea, nausea, vomiting), a profile similar to reports in conference abstract form for another CDK7 inhibitor in clinical development[16]. Gastrointestinal effects were generally low grade, reversible and ameliorated by standard anti-nausea and anti-diarrhea therapies. Future studies will monitor the gastrointestinal profile and will investigate the benefits of routine anti-emetic prophylaxis. In the longer term, a switch from the current dosing formulation of multiple instant-release capsules to a single tablet formulation is planned, a change anticipated to further enhance tolerability.

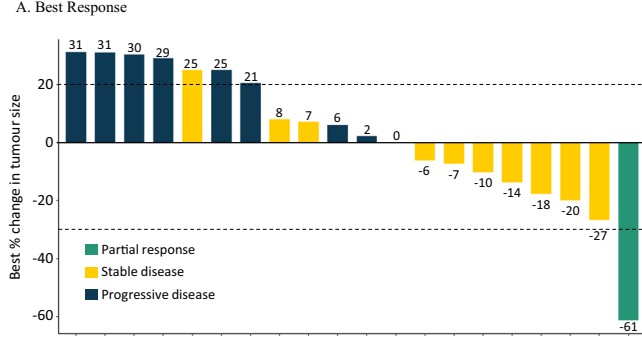

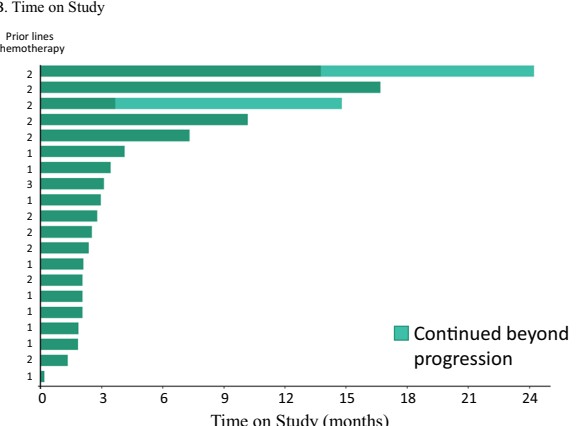

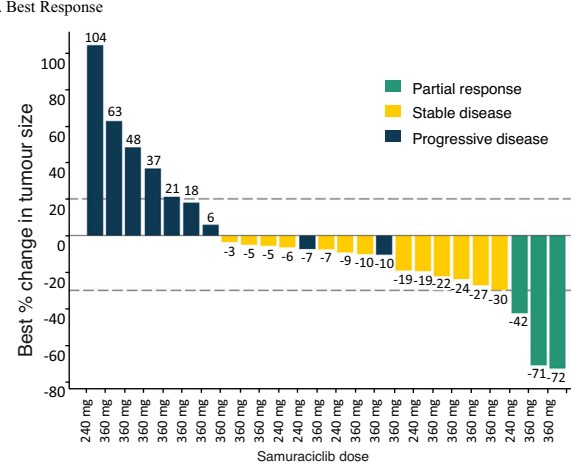

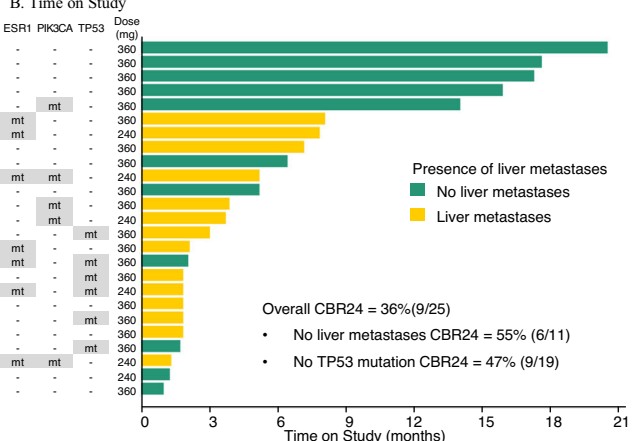

**Fig. 2 | Best RECIST response and time on study in Module 1B-1. A** Best response. Each bar represents the best percentage change from baseline for an individual patient. $N = 19$ (Note: only participants with at least one post-baseline value are included in this figure). * Lesion size increase from 12 to 15 mm was below the threshold for classification as progressive disease (minimum increase of 5 mm). **B** Time on study. Time on study for each patient (prior lines of chemotherapy for each patient indicated). $N = 20$.

**Fig. 3 | Best RECIST response and time on study in Module 2A. A** Best response. Each bar represents the best percentage change from baseline for an individual patient. N = 25 (Note: only participants with at least one post-baseline value are included in this figure). **B** Time on study. Time on study for each patient (presence of liver metastases and mutation status for each patient also indicated). $N = 25$. CBR24 clinical benefit rate at 24 weeks, mt mutation detected.

Thrombocytopenia was reported as Grade ≥4 in 3 patients (1 with predisposing factors), but this was rapidly reversed after discontinuation of samuraciclib. A mild drop in platelets was seen across the cohorts and is likely to be an on-target consequence of CDK7 inhibition given that increased activity of CDK7 in megakaryocytes has been associated with their maturation[17]. Neither neutropenia nor alopecia were observed even at the highest dose of samuraciclib.

PK analysis showed that samuraciclib is orally available, with exposure being dose-proportional and time independent. The relatively long half-life supports OD dosing. No clinical PK interactions have been found to date.

In Module 1A, initial evidence of anti-tumor activity was demonstrated by a DCR of 52.8%, predominantly consisting of SD across a range of cancer types, with a notable benefit seen in CRPC. CRPC was predicted to be sensitive to samuraciclib due, potentially, to activation of androgen receptors by CDK7, which has been shown to occur in preclinical studies, in an analogous fashion to the reduction of estrogen receptor (ER) activity by CDK7 inhibitors in breast cancer[3,18]. Previous pre-clinical studies using THZ1 (which inhibits CDK12 and CDK13 as well as CDK7) have predicted benefit in other cancer types in which enhanced transcription is a major feature of the cancer phenotype[19], giving scope for samuraciclib use in a wide range of cancers[6,19–22].

In the Module 1B-1 TNBC expansion, evidence of anti-tumor activity was demonstrated by a CBR of 20.0%, including 1 PR. Previous preclinical studies have suggested that enriching the TNBC patient population for SOX9, which interacts with FOXC1 to activate MYC may result in an improved DCR[12]. Other studies have shown a potential role for combining CDK7 inhibition with BH3 mimetics[6] and inhibitors of TGFbeta/Activin which is responsible for up-regulating ABCG2 and potentially ABCB1 transporter, responsible for resistance to CDK7 inhibitors[7,13]. Although treatment of TNBC has recently improved with the introduction of PARP inhibitors and immunomodulators, these treatments are only temporarily effective and there may be a role for combining inhibitors of transcription such as CDK7 inhibitors by analogy with the suggested combinations of PARP inhibitors with CDK12 inhibitors[23]. It is possible that, since we rarely encountered hematological adverse effects, samuraciclib could be added to cytotoxic chemotherapy or other treatments for TNBC in the future.

Selective estrogen receptor degraders (SERDs) such as fulvestrant have limited activity after progression on CDK4/6 inhibitors when given as a single agent[24–27]. The results from Module 2A suggest the combination of fulvestrant with samuraciclib may provide clinically meaningful activity, with a median PFS of 3.7 months in the intent-to-treat (ITT) population, double that expected for fulvestrant alone[15,25,28]. In a pre-planned exploratory analysis, PFS was 7.4 months in univariate analysis in patients with no TP53 mutation detected in ctDNA at baseline. Previous studies have shown that approximately 70% of patients with metastatic breast cancer do not have TP53 mutations

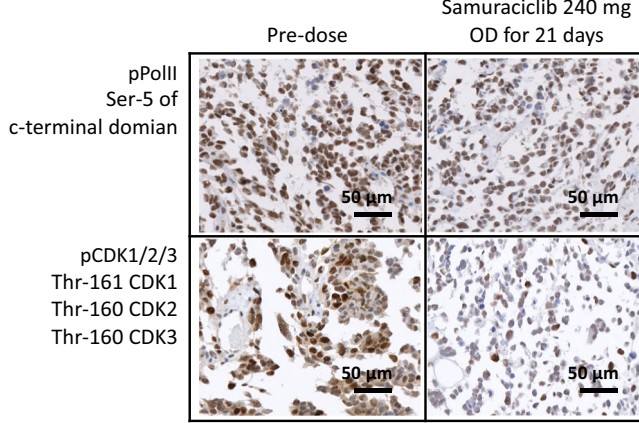

**Fig. 4 | Evidence of target engagement.** Reductions in CDK1/2/3 and pPolII levels in paired tumor biopsy samples demonstrates target engagement and proof of mechanism. Images are representative of two technical replicates from paired biopsies from a single patient. See Supplementary Table 11 for quantification and additional data from a further 5 patient paired biopsies. CDK cyclin-dependent kinase, OD once daily, pCDK phosphorylated cyclin-dependent kinase, pPolII phosphorylated RNA polymerase II.

disclosed by ctDNA analysis. The impact of lack of TP53 mutation on PFS in Module 2A was greater than that observed previously for fulvestrant alone or in combination with palbociclib[29,30], which argues against this being a purely prognostic finding and supports an interaction between samuraciclib treatment and TP53 status. Indeed, nonclinical data with BS181, an analog of samuraciclib, indicated that its activity was in part dependent on TP53 function[10,11]. Our ongoing work indicates that CDK7 inhibition can enhance the transcriptional activity of TP53, as measured by TP53 regulated gene expression, and that this is particularly evident in HR+ breast cancer cell lines. Consequently, the predictive potential of TP53 status for samuraciclib in combination with SERD therapy will be prospectively evaluated in multivariate analyses in the future. The potential benefit in patients with absence of liver metastases at the time of progression on CDK4/6i inhibitor therapy will also be evaluated prospectively[31]. No significant relationship between ESR1 mutations in ctDNA and response to samuraciclib was seen, therefore ESR1 mutation-positive patients could also potentially benefit from samuraciclib therapy.

If TP53 status is borne out to be a predictive biomarker in future trials, the potential to achieve this magnitude of benefit is important due to the paucity of effective targeted treatments after resistance to endocrine therapy plus CDK4/6 inhibition. Unfortunately, there is currently no consensus regarding standard of care for the treatment of women with advanced HR+/HER2− breast cancer whose disease progresses on CDK4/6 inhibition. While patients are frequently switched to cytotoxic agents such as paclitaxel and capecitabine, if combination endocrine treatment is continued options include the mTOR inhibitor everolimus and the PI3K inhibitor alpelisib. A non-randomized retrospective study of everolimus reported a median duration of treatment of 3.6 months, plus tolerability issues such as stomatitis and pneumonitis[26]. Alpelisib is limited to patients with a PIK3CA mutation (approximately 40% of patients) and in a single arm post CDK4/6 inhibitor cohort achieved a median PFS of 7.3 months, but many patients need anti-diabetic medication due to hyperglycemia[27]. The potential benefit of combined fulvestrant and samuraciclib is therefore important in its own right, but also because new oral SERDs are currently in clinical trials and may be more effective partner therapies than fulvestrant. The efficacy signal of samuraciclib therefore may be further increased in combination with the emerging therapeutic class of oral SERDs[25].

The PK and PD of the 2 doses of 240 mg and 360 mg were reviewed by the Safety Review Committee (SRC) and a starting dose of 360 mg OD appeared to be the most promising dose level to start treatment based on PK and PD data. However, this is based on the understanding that a dose reduction to 240 mg could be considered if the 360 mg dose is not tolerated. In line with current best practice (Project OPTIMUS, FDA) further data to evaluate both the 240 mg and 360 mg doses are being collected in new Phase 2 trials.

Overall, the data show an acceptable safety profile and initial evidence of activity for samuraciclib as a selective inhibitor of CDK7. Limitations of this dataset are that the studies were uncontrolled, nonrandomized, single arm evaluations in a relatively small number of patients (TNBC cohort). Thus, in summary, samuraciclib has the potential to address the significant medical need of patients whose disease has progressed on CDK4/6 inhibitors and this will be investigated further in future studies. A number of co-administration studies with both fulvestrant and new oral SERDs are now being initiated. Additionally, further studies are warranted to evaluate which other cancer types and combination strategies are most promising for samuraciclib therapy.

## Methods
The study was approved by the regulatory authorities, the Yorkshire & The Humber - Leeds West Research Ethics Committee, Jarrow, UK and the local ethics committees for each site. The study was conducted in accordance with ICH-GCP guidelines, the Declaration of Helsinki and all legal, regulatory, and data protection requirements. All patients provided written informed consent prior to participation.

Patients were recruited at sites in the UK and the USA by suitably qualified investigators with experience in oncology clinical trials. Patient enrollment dates were from 14 November 2017 to 07 May 2020 in Module 1A, from 19 January 2019 to 14 May 2021 in Module 1B-1, and from 12 November 2019 to 04 April 2022 (cut-off date for data analysis) in Module 2A. Patients were recruited in a non-randomized open-label fashion based on the investigator's pool of potential patients. Patients were then screened against the inclusion/exclusion criteria and included if eligible.

All data were collected using IBM Clinical Development Version 2019.3.0.1 and SAS System Version 9.1 was used for the data analysis.

### Module 1A
In this Phase I study (ClinicalTrials.gov: NCT03363893), male and female patients with locally advanced/metastatic solid tumors received samuraciclib in dose escalation cohorts (at least 3 and up to 6 evaluable patients permitted per dose cohort) to determine the MTD (see Supplementary Information for definition). An additional 6 to 12 male or female patients with breast cancer were enrolled to a paired biopsy cohort to evaluate the pharmacodynamic effects of samuraciclib in tumor tissue, initiated after determining the minimally biologically active dose, defined by a 25-30% reduction in signal for pPolII in lymphocytes after 21 days of dosing or more.

Patients aged 18 or over, ECOG performance status 0 or 1, estimated life expectancy of >12 weeks, and histological, radiological, or cytological confirmation of advanced non-hematological malignancy not considered appropriate for further standard treatment were eligible. For inclusion in the paired biopsy cohort, only patients with breast cancer with lesions amenable to biopsy were recruited. A full list of inclusion/exclusion criteria is provided in the Supplementary Information.

Patients initially received an oral dose of samuraciclib at Cycle 0, Day 1 for PK evaluation, followed by a 48-hour interval then continuous daily dosing for 21 days (Cycle 1). Dosing continued in 21-day cycles until the patient no longer gained clinical benefit or had intolerable toxicity. At least 3 evaluable patients in a cohort had to complete Cycle 1 before any dose escalation was considered.

The primary endpoint was safety and tolerability, with secondary endpoints to characterize the PK of samuraciclib and to assess biological and anti-tumor activity.

## Module 1B-1

A single arm expansion was conducted to refine the safety, tolerability, PK, and pharmacodynamic profiles of samuraciclib monotherapy (360 mg OD) in male or female patients with advanced TNBC. Patients with histological, radiological or cytological confirmation of metastasis or locally advanced TNBC not considered to be appropriate for further standard treatment, who had received at least 1 line of systemic anti-cancer therapy, and measurable disease according to the RECIST V1.1, were eligible.

Patients underwent regular safety monitoring and 8-week scans reported using RECIST V1.1.

## Module 2A

A single-arm, ascending-dose study was conducted in 2 cohorts to explore the recommended dose of samuraciclib and fulvestrant in HR+/HER2− advanced breast cancer patients who had previously received, and become resistant to, a CDK4/6 inhibitor. Female patients with histologically confirmed diagnosis of breast cancer, evidence of metastatic or locally advanced disease, and documented ER+ and/or PgR+ and HER2- tumor status treated with an AI in combination with a CDK4/6i before study entry were eligible. In each cohort the dose of fulvestrant was 500 mg every $28 \pm 2$ days, with an additional 500 mg dose given $14 \pm 2$ days after the first dose. Cohort 1 tested samuraciclib 240 mg OD continuous dosing. If the DLT stopping criteria were not met, Cohort 2 commenced enrollment at the Module 1A MTD. Patients underwent regular safety monitoring and 8-week scans reported using RECIST V1.1[32].

## End points

As Module 1A was a 'first time in human' safety study, initial efficacy was evaluated as a secondary endpoint using the DCR, defined as the percentage of patients with a complete response (CR), PR or stabilization of disease at first on treatment RECIST assessment. To allow for the evaluation of longer-term benefit, Module 1B-1 and Module 2A used the CBR as a secondary endpoint, defined as the percentage of patients with CR or PR or stabilization of disease for at least 24 weeks between enrollment and disease progression or death due to any cause. In addition, the ORR was assessed, defined as the percentage of patients who had at least 1 objective response (CR or PR) prior to any evidence of progression. RECIST V1.1 endpoints were assessed in the response evaluable population - defined as all patients who received ≥1 dose of samuraciclib and had a post-baseline tumor assessment. Statistical analyses of PFS using the Kaplan-Meier method were performed on the ITT population - defined as all enrolled patients. AEs were summarized from the Safety Population - defined as all patients who received at least 1 dose of samuraciclib - using the MedDRA system organ class (SOC), preferred term (PT), and graded according to CTCAE V5.0 (Supplementary Information).

## Pharmacokinetic/pharmacodynamic analyses

PK samples were taken in Module 1A, Module 1 B-1 and Module 2A (see schedule of assessments in Supplementary Information for timing of PK samples). PK parameters were derived using standard non-compartmental methods.

Per-protocol ctDNA samples were taken in Module 1B-1 at screening, C1D1, C2D1 and on D1 of alternating cycles from Cycle 3; in Module 2A at screening, C1D1 and at either C2D1 (240 mg) or C2D15 (360 mg) for mutational analysis[24,33], (Supplementary Information). pPolII in lymphocytes and pPolII and pCDK1/2/3 in core needle tissue samples from patients recruited to the paired biopsy cohort were evaluated for on-target pharmacodynamic analyses.

## Reporting summary

Further information on research design is available in the Nature Portfolio Reporting Summary linked to this article.

## Data availability

The individual participant data from this study cannot be made publicly available due to the sponsor, Carrick Therapeutics, contractual obligations. Data may be requested after the product and indication has been approved by major health authorities and/or 24 months after completion of all the clinical study reports for the reported arms of the NCT03363893 trial (anticipated to be completed by the end of 2023). Researchers should submit a proposal to the corresponding author (matthew.krebs@manchester.ac.uk) outlining the reasons for requiring the data. Applications should specifically outline the data the parties are interested in receiving and how the data will be used; the use of the data must also comply with the country- or region-specific regulations and will be supplied as de-identified data so individual participants cannot be identified. The corresponding author and sponsor will endeavor to respond to requests within 6 weeks of receipt. A signed data access agreement with the sponsor is required before accessing the shared data and access will be limited to a defined period (to be agreed with the requestor). Study protocols (core protocol plus Module 1B-1 and Module 2 protocols) are included in the Supplementary Information file. The remaining data are available within the article and its Supplementary Information.

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

## Acknowledgements

The authors would like to acknowledge the input from the following people: Ed Ainscow, Anthony Barrett, Paul Dickinson, Iain Macpherson, Kristine Pemberton and Manfred Lehnert. The study was designed and funded by Carrick Therapeutics. Carrick Therapeutics also provided funding to third party Contract Research Organisations for data collection and processing. Debbie Jordan Ltd provided medical writing services for this manuscript, with the cost of these services paid by Carrick Therapeutics. R.C.C. and S.A. would like to acknowledge Cancer Research UK for funding (C37/A18784). M.G.K. and S.H. acknowledge support from National Institute for Health Research (NIHR) Manchester Biomedical Research Centre, NIHR Manchester Clinical Research Facility at The Christie and Manchester Experimental Cancer Medicine Centre (Manchester, UK). S.R.L. acknowledges support from The Oxford Experimental Cancer Medicine Centre and National Institute for Health Research (NIHR) Oxford Biomedical Research Centre. L.K. acknowledges infrastructure support from the Imperial Experimental Cancer Medicine Centre, Cancer Research UK Imperial Centre, National Institute for Health Research (NIHR), Imperial Biomedical Research Centre (BRC), Imperial College Healthcare NHS Trust Tissue Bank, and the NIHR Imperial Clinical Research Facility. C.P. acknowledges support from The Liverpool Experimental Cancer Medicine Centre [Grant Reference: C18616/A25153], Cancer Research UK. R.B. acknowledges support from the Cancer Research UK Cambridge Centre, Cambridge Experimental Cancer Medicine Centre and NIHR Cambridge Biomedical Research Centre including Cambridge Clinical Research Centre (grant #BRC-1215-20014). The views expressed in this paper are those of the authors and not necessarily those of the supporting bodies listed above.

## Author contributions

These authors jointly supervised this work (as recruiting investigators and being involved in study design and data interpretation): M.K., R.C.C., S.H., S.L., and R.J. These authors contributed equally (as recruiting investigators): L.K., J.Ma., Z.M., C.P., L.C., P.R., W.G., S.S., J.Me., J.O., P.W., P.C., T.A., R.B. The following author was a preclinical scientist involved in data interpretation: S.A. The following authors are Carrick employees and were involved in the study design and data interpretation: G.C., A.B., S.M.

## Competing interests

R.C.C. has had a travel grant from Carrick Therapeutics and owns a patent on samuraciclib (patent number WO/2015/124941); he also has a grant from AstraZeneca. S.H. has received speaker fees Pfizer and advisory board and grant funding from Lilly. S.R.L. has received consulting fees from Sanofi, GLG consulting, Atheneum and Rejuversen. He has also received payment or honoraria for lectures, presentations, or educational events from Eisai, Prosigna, Roche, Pfizer, Novartis, Shionogi and Sanofi, and was previously employed by Pfizer. He has received travel, accommodation or expenses from Pfizer, Roche, Synthon and Piqur Therapeutics, and research funding from CRUK, Against Breast Cancer, Pathios Therapeutics and is cofounder of Mitox Therapeutics. His institution has received funding for clinical trials for which he was chief/principal investigator from CRUK, Boehringer Ingelheim, Piqur Therapeutics, Astra Zeneca, Carrick Therapeutics, Sanofi, Merck KGaA, Synthon, Roche and Prostate Cancer UK. J.O. has received honoraria for consulting and/or advisory boards from the following: AbbVie Inc., Agendia, Amgen Biotechnology, Aptitude Health, AstraZeneca, Bayer, Bristol-Myers Squibb, Carrick Therapeutics, Celgene Corporation, Clovis Oncology, Daiichi Sankyo, Eisai, G1 Therapeutics, Genentech, Gilead Sciences, GRAIL, Halozyme Therapeutics, Heron Therapeutics, Immunomedics, Ipsen Biopharmaceuticals, Lilly, Merck, Myriad, Nektar

Therapeutics, Novartis, Ontada, Pfizer, Pharmacyclics, Pierre Fabre Pharmaceuticals, Puma Biotechnology, Prime Oncology, Roche, Samsung Bioepis, Sanofi, Seagen, Syndax Pharmaceuticals, Taiho Oncology, Takeda, and Synthon. ZM has a consulting/advisory role with AstraZeneca, Gilead Sciences and Daiichi Sankyo. He has also received research funding for his institution from Seattle Genetics, Novartis, AstraZeneca, Radius Health, Daiichi Sankyo, Lilly, GlaxoSmithKline, and Olema Oncology. CP acknowledges support from The Liverpool Experimental Cancer Medicine Centre [Grant Reference:C18616/A25153], The Clatterbridge Cancer Charity, North West Cancer and Make 2nds Count. CP reports grant funding support from Pfizer, Daiichi Sankyo, Exact Sciences, Gilead and Seagen. Honoraria for advisory boards have been received from Pfizer, Roche, Daiichi Sankyo, Novartis, Exact sciences, Gilead, SeaGen and Eli Lilly and support for travel and conferences from Roche, Novartis and Gilead. T.A. is employed by Sarah Cannon/HCA Healthcare UK and Ellipses Pharma and receives funding in an advisory capacity from iOnctura, Labgenius, and Servier. R.D.B. has a consulting/advisory role with Shionogi, Daiichi Sankyo, Molecular Partners, Roche/Genentech, Novartis, and AstraZeneca. He has also received research funding for his institution from AstraZeneca, Genentech, Shionogi, Molecular Partners, Sanofi, Boehringer Ingelheim, Roche, Biomarin, GI Therapeutics, and Carrick Therapeutics, and travel/accommodation/expenses from Shionogi, AstraZeneca, Molecular Partners, and Daiichi Sankyo. R.J. has received research funding from Pfizer and Lilly, and she serves on an advisory board for Carrick Therapeutics and Luminex. S.A. is a named inventor on patents describing CDK inhibitors, including samuraciclib, owns shares in Carrick Therapeutics and has been funded by Carrick Therapeutics; he also has grants from AstraZeneca. G.C., A.B., and S.M. are Carrick employees and shareholders. M.K. has received honoraria from Janssen, Roche; consulting/advisory fees from Achilles Therapeutics, Bayer, Guardant Health, Janssen, OM Pharma, Roche, Seattle Genetics; speakers fees from AstraZeneca, Janssen, Roche and research funding from Roche and Novartis for his institution. He has received travel, accommodation or expenses from AstraZeneca, BerGenBio, Immutep, Janssen and Roche. The remaining authors declare no competing interests.

## Additional information

R. C. Coombes [1], Sacha Howell[2], Simon R. Lord [3], Laura Kenny[1], Janine Mansi[4], Zahi Mitri [5], Carlo Palmieri [6], Linnea I. Chap[7], Paul Richards[8], William Gradishar [9], Sagar Sardesai[10], Jason Melear[11], Joyce O'Shaughnessy[11], Patrick Ward [10], Pavani Chalasani[12], Tobias Arkenau[13], Richard D. Baird[14], Rinath Jeselsohn [15], Simak Ali [1], Glen Clack[16], Ashwani Bahl[16], Stuart McIntosh[16] & Matthew G. Krebs [2] ✉

[1]Imperial College, South Kensington, London, UK. [2]Division of Cancer Sciences, Faculty of Biology, Medicine and Health, The University of Manchester and The Christie NHS Foundation Trust, Manchester Academic Health Science Centre, Manchester, UK. [3]Early Phase Clinical Trials Unit, Department of Oncology, University of Oxford, Oxford, UK. [4]Guy's and St Thomas' NHS Foundation Trust, London, UK. [5]OHSU Knight Cancer Institute, Portland, OR, USA. [6]University of Liverpool, Liverpool, UK. [7]University of California, Los Angeles, CA, USA. [8]Blue Ridge Cancer Center, Salem, VA, USA. [9]Northwestern University, Chicago, IL, USA. [10]US Oncology Research, OHC, Cincinnati, OH, USA. [11]Baylor University Medical Center, Texas Oncology, Dallas, TX, USA. [12]University of Arizona Cancer Center, Tucson, AZ, USA. [13]Sarah Cannon Research Institute, London, UK. [14]Cancer Research UK Cambridge Centre, Cambridge, UK. [15]Dana-Farber Cancer Institute, Boston, MA, USA. [16]Carrick Therapeutics, Dublin, Ireland. ✉e-mail: matthew.krebs@manchester.ac.uk

