## [Peer Review File · Nature Communications]

DOSE ESCALATION AND EXPANSION COHORTS IN PATIENTS WITH ADVANCED BREAST CANCER IN A PHASE I STUDY OF THE CDK7-INHIBITOR SAMURACICLIBREVIEWER COMMENTS

Reviewer #1 (Remarks to the Author): with expertise in medical oncology, early phase clinical trials

In this manuscript, Coombes and colleagues present the results of a Phase I dose escalation clinical trial of the novel, first in class reported, CDK7-inhibitor, samuraciclib, in advanced solid tumor patients with 11 patients with breast cancer added who underwent serial biopsies and a monotherapy cohort in TNBC (Module 1A). There are also a cohort in patients with HR+ breast cancer who had failed a CDK 4/6 combination (Module 2A). This cohort combined samuaraciclib with fulvestrant : 6 patients treated with a 240mg daily samuaciclib dose and 25 patients with a 360mg daily samuaciclib dose. The manuscript is well written and easy to follow. This is the first report of a CDK 7 inhibitor in clinical investigation and specifically first report of samuraciclib in clinical investigation. Although all components of the trial were "single arm," there did appear to be clinical benefit in all modules investigated.

The reviewer has the following questions for the authors:

- 1) In some modules stable disease is defined as 24 weeks minimum. However, it is unclear if this definition of stable disease also relates to that discussed in Module A, lines 152-154. Please define consistently throughout all modules.
- 2) Lines 210-211. The authors discuss 6 pre-menopausal patients who were treated in Module 2A. Were these patients chemically post-menopausal?
- 3) Can the authors comment on PD effects in the trial? It did not appear, as was briefly discussed in the text, that the PD effects were dose proportional - at least for 240mg and 360mg doses, the 2 doses of greatest interest for advancing the agent into further clinical trials. If there is no dose-proportional increase in PD effects, the reviewer is concerned about the decision to advance the 360mg dose into subsequent Phase 2/3 trials as is recommended
- 4) In paragraph beginning on line 230, specifically sentence beginning on line 235 that describes toxicities associated with combination therapy with fulvestrant - the reviewer is concerned that 23% of patients discontinued treatment due to GI AEs. Even though they were grade 2/3 - this is still significant. It is difficult for patients to tolerate GI toxicities long term that are grade 2/3. Can the authors break this down here into specific doses for which the toxicities occurred? There is concern that the wrong dose of samuraciclib may be advancing forward in future combination strategies. Also can the authors be more specific pursuant to the GI toxicities in this section and associated dose?
- 5) Line 241: There were 5 deaths on trial - although these were all considered disease related, the reviewer is concerned about patient selection. This must be watched carefully in future clinical trials.
- 6) Line 246: There were 6/27 drug related SAEs. Can the authors indicate in text doses for which these occurred to help better understand toxicity/safety profile of the chosen Recommended Phase 2 Dose (RP2D)?
- 7) Line 257-259: Can the authors comment on the duration of time to recovery of the thrombocytopenia and at what doses the Grade 3+ thrombocytopenia occurred? As there are only 2 patients, can the authors comment specifically on this toxicity?
- 8) Lines 283-285: The authors indicate that there were 3 PR and 13 SD in Module 2A of 25 patients. They are then saying that the CBR was 36% or 9/25 patients. If SD is considered CBR, can the authors identify the 9 patients as oppose to 13 (SD) + 3 (PR) = 16 patients?
- 9) Can the authors comment on % variability in bioavailability amongst patients within each dosing cohort? Also was a component of food effect done to know if the drug has significant variability predicated on dietary status? This would be very important information to know within the context of

this paper and also in determining dose chosen for further investigation.

Reviewer #2 (Remarks to the Author): with expertise in breast cancer, clinical

The authors report results of a phase I study of novel CDK7 inhibitor samuraciclib.

There is a strong preclinical rationale to develop this inhibitor. The design of the study, methods and endpoints are very well defined. The results are well presented and detailed.

In this manuscript the results of the first modules are presented:

- 1A first in man, all tumors with PD analysis
- 1B-1 single arm expansion in TNBC at the dosage of 360 mg
- 2A single arm HR+ HER2- in combination with fulvestrant

In the module 1Aa, in a paired biopsy cohort, the pharmacodynamics effects of samuraciclib was evaluated with evidence of target engagement as assessed with pPoLL and pCDK1/2.

The main toxicities are mainly GI with diarrhea, nausea, vomiting with high incidence, even if few grade 3 (less than 10%). The authors state that GI effects are low grade, reversible and ameliorated by standard anti-nausea and anti-diarrhea treatments. It would be of interest to know the duration of such events and the repartition between grade 1 and 2.

The Maximum tolerated dose was 360 mg once daily with target engagement ≥ 120 mg.

With the limits of phase I trial, small number of patients with high heterogeneity of population and heavily pretreated patients, clearly there is hint of preliminary activity deserving further evaluation of this compound. Interestingly TP53 status should be evaluated in future trials as predictive biomarker. As Stated by the authors, it is the first report of CDK7 inhibitor in human and data strongly suggest high level of interest in future development.

Reviewer #3 (Remarks to the Author): with expertise in biostatistics, clinical trial study design

The study details three study cohorts (1A, 1B and 2A) with endpoints (DRC, ORR, CBR, and PFS). Dose evaluation for identifying DLT, AE report, PK, PD, and efficacy analysis were also described clearly.

Statistical comments for clarification:

1. Information blocked in the clinical trial protocol. There are many places blocked in the protocol which make difficult for evaluation.
2. The patient numbers in the protocol (Table 1) are not matched to the manuscript (e.g., paired samples: $n=9$ for 240 mg + $n=9$ for 360mg in the protocol's Table 1 vs $n=11$ in the manuscript's Table 1).
3. Need stratification of cancer type in Cohort 1A dose escalation. The manuscript focuses on breast cancer. However, Cohort 1A dose escalation is a mixture of breast cancer (30%) with other cancer types. Report of cohort 1A should be stratified by at least breast cancer vs non-BC for demographic, AE, and efficacy in text, table, and figure.
4. Figure 2 of Cohort 1B may need a boxplot with t-test or Wilcoxon test to easily visualize and test difference of treatment duration for number of prior therapy (1 vs >1). The same strategy could be also applied to Figure 3 for liver met vs non-liver met, as well as mutation (e.g., TP53)
5. It would be more informative to generate KM survival plot overall, liver met status, TP53 status for cohort 2A in addition to Table 3.
6. Is there any association of patients experiencing only low-grade AEs with clinical outcomes or biomarker (e.g., PFS, treatment duration, or response)?

Reviewer #4 (Remarks to the Author): with expertise in breast cancer, CDK7

CDK7 inhibition is a promising therapeutic strategy in cancer. Samuraciclib (CT7001) as a small molecule, ATP competitive, selective oral inhibitor of CDK 7 was firstly reported in ESMO in 2021, it has shown an acceptable safety profile with evidence of anti-tumour activity in combination with fulvestrant for patients with advanced HR+BC who have progressed on their prior CDK4/6i. The authors of this this manuscript extend the evaluation of safety and anti-tumor activity of samuraciclib in the subgroup of TNBC. The authors demonstrate the special association between TP53 and samuraciclib. It would be helpful to include the further discussion of the underlying mechanism between TP53 and CDK7. Moreover, TP53 is mutated in about 80% of TNBC and ESR2 is expressed in approximately in 60-80% TNBC (Key paper in this area is PMID 30990221), it may be helpful to provide the data of TP53 and CDK7 in the subgroup of TNBC. Some limitation of the analysis should not be omitted, as the nonrandomized with a relatively small number of TNBC, and there was a lack of prospective HER2 assessment description by a central laboratory additionally.

Point-by-point response to the reviewers' comments

Reviewer #1 (Remarks to the Author): with expertise in medical oncology, early phase clinical trials

In this manuscript, Coombes and colleagues present the results of a Phase I dose escalation clinical trial of the novel, first in class reported, CDK7-inhibitor, samuraciclib, in advanced solid tumor patients with 11 patients with breast cancer added who underwent serial biopsies and a monotherapy cohort in TNBC (Module 1A). There are also a cohort in patients with HR+ breast cancer who had failed a CDK 4/6 combination (Module 2A). This cohort combined samuaraciclib with fulvestrant : 6 patients treated with a 240mg daily samuaciclib dose and 25 patients with a 360mg daily samuaciclib dose. The manuscript is well written and easy to follow. This is the first report of a CDK 7 inhibitor in clinical investigation and specifically first report of samuraciclib in clinical investigation. Although all components of the trial were "single arm," there did appear to be clinical benefit in all modules investigated.

The reviewer has the following questions for the authors:

1) In some modules stable disease is defined as 24 weeks minimum. However, it is unclear if this definition of stable disease also relates to that discussed in Module A, lines 152-154. Please define consistently throughout all modules.

Authors response: In Module 1A, a first-time-in-human solid tumor dose escalation study, disease control rate (DCR) is used as a secondary endpoint, defined as the percentage of participants with a CR, PR or SD at the first on treatment RECIST assessment. This aligns with the definition previously used for the covalent CDK7 inhibitor SY-1365 (Juric et al., Abstract#11,30th EORTC-NCI-AACR Symposium, Dublin 2018) and is used as a standard routine endpoint in Phase 1 trials. For modules 1B-1 and 2A, exploring activity in specific disease indications, CBR rate was used as a secondary endpoint to evaluate longer term benefit, defined as the percentage of patients with CR, PR or SD for at least 24 weeks. The text in the manuscript has been revised to make this clearer (End Points: Lines 153 to 163 in the track changes version) as follows:

As ~~in~~ Module 1A was a 'first time in human' safety study, initial efficacy was evaluated as a secondary endpoint using the disease control rate (DCR), defined as the percentage of participants with a complete response (CR), ~~or~~ partial response (PR) or stabilization of disease at first on treatment RECIST assessment. To allow for the evaluation of longer-term benefit, ~~in~~ Modules 1B-1 and Module 2A used the Objective Response Rate (ORR) was defined as the percentage of participants who had at least 1 objective response (CR or PR) prior to any evidence of progression and the Clinical Benefit Rate (CBR) as a secondary endpoint, defined as the percentage of patients with CR or PR or stabilization of disease for at least 24 weeks between enrolment and disease progression or death due to any cause. In addition, the Objective Response Rate (ORR) was assessed, defined as the percentage of participants who had at least 1 objective response (CR or PR) prior to any evidence of progression.

2) Lines 210-211. The authors discuss 6 pre-menopausal patients who were treated in Module 2A. Were these patients chemically post-menopausal?

Authors response: Yes, all 6 pre-menopausal patients were receiving goserelin. All 6 patients had already progressed on prior combined goserelin+aromatase inhibitor treatment, as required by the protocol inclusion criteria, and were not receiving any other concomitant anti-tumor therapy during this study, indicating any response in these patients would have been due to samuraciclib treatment. None of these patients were in fact responders. The text has been updated in the manuscript to reflect this (Demography and Baseline Characteristics: Lines 215 to 217) as follows:

In Module 2A, *the mean age was 60.4 years, and all patients were female. All patients had received prior AI in combination with CDK4/6. Six of 31 patients were pre-menopausal (all were receiving goserelin).*

3) Can the authors comment on PD effects in the trial? It did not appear, as was briefly discussed in the text, that the PD effects were dose proportional - at least for 240mg and 360mg doses, the 2 doses of greatest interest for advancing the agent into further clinical trials. If there is no dose-proportional increase in PD effects, the reviewer is concerned about the decision to advance the 360mg dose into subsequent Phase 2/3 trials as is recommended

Authors response: We thank the reviewer for this observation and comment. We agree that both doses appear effective and tolerated. A starting dose of 360mg OD was selected by the Safety Review Committee (SRC) to be the most promising dose level to take forward based on the available safety, pharmacokinetic and pharmacodynamic data. A dose reduction to 240mg was recommended at the discretion of the investigator if adverse events were not tolerable. In line with current best practice (Project OPTIMUS, FDA) further data to evaluate both the 240mg and 360mg doses are being collected in new Phase 2 trials. The following information has been added to the manuscript (Discussion: Lines 436 to 441 in the track changes version):

The PK and PD of the 2 doses of 240mg and 360mg were reviewed by the Safety Review Committee (SRC) and a starting dose of 360mg OD appeared to be the most promising dose level to start treatment based on PK and PD data. However, this is based on the understanding that a dose reduction to 240mg could be considered if the 360mg dose is not tolerated. In line with current best practice (Project OPTIMUS, FDA) further data to evaluate both the 240mg and 360mg doses are being collected in new Phase 2 trials.

4) In paragraph beginning on line 230, specifically sentence beginning on line 235 that describes toxicities associated with combination therapy with fulvestrant - the reviewer is concerned that 23% of patients discontinued treatment due to GI AEs. Even though they were grade 2/3 - this is still significant. It is difficult for patients to tolerate GI toxicities long term that are grade 2/3. Can the authors break this down here into specific doses for which the toxicities occurred? There is concern that the wrong dose of samuraciclib may be advancing forward in future combination strategies. Also can the authors be more specific pursuant to the GI toxicities in this section and associated dose?

Authors response: We thank the reviewer for this important point. GI events are a common finding for kinase inhibitors across multiple mechanisms e.g. CDK4/6, HER family including EGFR etc. The authors acknowledge such events can be challenging for patients but also note the successful use of these kinase inhibitor therapies in clinical practice. We would like to highlight that the data presented in this manuscript are from early studies with samuraciclib, during which the emerging safety profile was becoming understood and nausea prophylaxis was not initially permitted in order to accurately determine the safety profile of the compound. Treatment with nausea prophylaxis is now routinely used at the initiation of dosing and this appears to be a successful strategy in reducing the incidence of GI toxicity. This information is provided in Discussion: Lines 357 to 360 in the track changes version, and reads as follows:

Gastrointestinal effects were generally low grade, reversible and ameliorated by standard anti-nausea and anti-diarrhea therapies. Future studies will monitor the gastrointestinal profile and will investigate the benefits of routine anti-emetic prophylaxis.

Also can the authors be more specific pursuant to the GI toxicities in this section and associated dose?

Authors response: Table S10 in the supplementary information shows a breakdown of the AEs by 240mg and 360mg dose. There is no obvious dose-related increase in AEs between the two dose levels. As described above, other studies are ongoing at both dose levels and the totality of the AE, efficacy and PD data will be reviewed when the ongoing studies have completed. The following new text has been added to the manuscript:

- Safety Results: Lines 232 to 235 in the track changes version: *No difference in the AE profile was seen between the 240mg dose and the 360mg dose in combination with fulvestrant, and no obvious dose-related trends were seen (Table S10 Supplementary Information)*
- Safety Results: Lines 247 and 248 in the track changes version: *The discontinuations were similar in the 2 dose groups.*

5) Line 241: There were 5 deaths on trial - although these were all considered disease related, the reviewer is concerned about patient selection. This must be watched carefully in future clinical trials.

Authors response: The authors acknowledge this point but would observe that these disease related deaths occurred in a patient population with very advanced disease. The rates of treatment-related death are not outside expectation for such a population in early phase trials. The authors will of course continue to carefully select suitable patients and evaluate rate and causal relationship of deaths in all studies.

6) Line 246: There were 6/27 drug related SAEs. Can the authors indicate in text doses for which these occurred to help better understand toxicity/safety profile of the chosen Recommended Phase 2 Dose (RP2D)?

Authors response: We acknowledge the relevance of adding this information and have edited the text accordingly (Safety: Lines 256 to 261 in the track changes version):

...6 considered related to study treatment (2 diarrhea [1 at 180mg BID dose and 1 at 240mg OD dose], 1 thrombocytopenia [180mg BID dose], 1 liver injury in a patient with documented liver metastases [240mg OD dose], and 1 esophagitis and gastroesophageal reflux disease [180mg BID dose] and one patient with 4 events - anemia, diarrhea, dyspnea at rest, and thrombocytopenia [360mg OD dose]). The 180mg BID dose regimen was ultimately declared non-tolerable and not explored further.

7) Line 257-259: Can the authors comment on the duration of time to recovery of the thrombocytopenia and at what doses the Grade 3+ thrombocytopenia occurred? As there are only 2 patients, can the authors comment specifically on this toxicity?

Authors response: We thank the reviewer for this query. For the majority of patients, a slight reduction was seen in platelet counts on therapy, with this being an expected PD effect. Figure S3 Supplementary Information shows the average platelet counts for all 3 modules. We had not previously included the 28 day follow-up platelet counts after the end of samuraciclib treatment, but have now updated Figure S3 accordingly to include this information. The data illustrate platelet counts were recovering at this time point. We have also updated the text in the manuscript as follows and included additional information regarding the two patients with G3 thrombocytopenia:

- Safety: Lines 267 to 269 in the track changes version: In the majority of patients, platelet counts remained within the normal range, and were not associated with bleeding events, and resolved when treatment ended.
- Safety: Lines 275 to 277 in the track changes version: *This finding of thrombocytopenia was subsequently confirmed in Module 1B-1 and Module 2A, with evidence of recovery within 28 days of stopping samuraciclib treatment (Figure S3 Supplementary Information).*

A summary of both patients with G3 toxicity has been added as follows (Safety: Lines 269 to 274 in the track changes version):

One patient, receiving samuraciclib 180mg BID, experienced a Grade 4 event that required transfusion with 1 unit of platelets. The patient had a pre-existing Grade 1 thrombocytopenia prior to dosing. A second patient receiving samuraciclib 360mg OD experienced Grade 3 thrombocytopenia. For both patients the events were resolving to Grade 1 by 11 days and 23 days, respectively, after samuraciclib dosing ended.

8) Lines 283-285: The authors indicate that there were 3 PR and 13 SD in Module 2A of 25 patients. They are then saying that the CBR was 36% or 9/25 patients. If SD is considered CBR, can the authors identify the 9 patients as opposed to 13 (SD) + 3 (PR) = 16 patients?

Authors response: We apologise for the lack of clarity around CBR and thank the reviewer for highlighting this. As detailed in our response to Question 1, we have updated the wording in the manuscript to better clarify the use of DCR and CBR (see below) In relation to this specific comment, while 3 patients achieved PR and 13 patients achieved SD in Module 2A as the best RECIST response at any timepoint during study participation, 2 patients with PR

and 7 patients with SD persisted for ≥ 24 weeks (which is the definition of CBR in Module 2A).

End Points: Lines 153 to 163 in the track changes version has been updated as follows:

As in Module 1A was a 'first time in human' safety study, initial efficacy was evaluated as a secondary endpoint using the disease control rate (DCR), defined as the percentage of participants with a complete response (CR), ~~or~~ partial response (PR) or stabilization of disease at first on treatment RECIST assessment. To allow for the evaluation of longer-term benefit, in Modules 1B-1 and Module 2A used the Objective Response Rate (ORR) was defined as the percentage of participants who had at least 1 objective response (CR or PR) prior to any evidence of progression and the Clinical Benefit Rate (CBR) as a secondary endpoint, defined as the percentage of patients with CR or PR or stabilization of disease for at least 24 weeks between enrolment and disease progression or death due to any cause. In addition, the Objective Response Rate (ORR) was assessed, defined as the percentage of participants who had at least 1 objective response (CR or PR) prior to any evidence of progression.

9) Can the authors comment on % variability in bioavailability amongst patients within each dosing cohort? Also was a component of food effect done to know if the drug has significant variability predicated on dietary status? This would be very important information to know within the context of this paper and also in determining dose chosen for further investigation.

Authors response: Figure S7 in the Supplementary Information shows the plasma PK profile of the various samuraciclib doses used in the modules, although the authors did not include error bars on this specific figure to facilitate clarity for the viewer. However, Figures S8 and S9 in the Supplementary Information shows the variability in the samuraciclib trough concentrations after dosing with 360mg, as monotherapy and in combination with fulvestrant, respectively. A food effect study has been undertaken and will be reported in a separate, dedicated, PK publication. In brief, no food effect was seen in relation to exposure or % variability.

Reviewer #2 (Remarks to the Author): with expertise in breast cancer, clinical

The authors report results of a phase I study of novel CDK7 inhibitor samuraciclib. There is a strong preclinical rationale to develop this inhibitor. The design of the study, methods and endpoints are very well defined. The results are well presented and detailed.

In this manuscript the results of the first modules are presented:

- 1A first in man , all tumors with PD analysis**
- 1B-1 single arm expansion in TNBC at the dosage of 360 mg**
- 2A single arm HR+ HER2- in combination with fulvestrant**

In the module 1Aa, in a paired biopsy cohort, the pharmacodynamics effects of samuraciclib was evaluated with evidence of target engagement as assessed with pPoLL and pCDK1/2.

The main toxicities are mainly GI with diarrhea, nausea, vomiting with high incidence, even if few grade 3 (less than 10%). The authors state that GI effects are low grade, reversible and ameliorated by standard anti-nausea and anti-diarrhea treatments. It would be of interest to know the duration of such events and the repartition between grade 1 and 2.

The Maximum tolerated dose was 360 mg once daily with target engagement ≥ 120 mg.

With the limits of phase I trial, small number of patients with high heterogeneity of population and heavily pretreated patients, clearly there is hint of preliminary activity deserving further evaluation of this compound. Interestingly TP53 status should be evaluated in future trials as predictive biomarker.

As Stated by the authors, it is the first report of CDK7 inhibitor in human and data strongly suggest high level of interest in future development.

Authors response: We thank Reviewer 2 for their kind and supportive summary of our data. We refer back to our response to Reviewer 1, Question 4 addressing a similar point in relation to GI toxicity.

Unfortunately, the data on duration of GI events was not collected consistently by all sites in these early studies and so provide limited information. The reason for this is the inconsistency between sites in applying stop dates to specific toxicities. For example, once nausea prophylaxis was prescribed and symptoms improved or resolved, the AE of nausea was often still left open on the AE reporting as a concomitant medication was ongoing. This makes interpretation of the actual duration of symptoms very challenging and is an issue shared across many early phase studies. This is being addressed in future studies of samuraciclib with both a management plan for GI events and a standardized way of recording duration of events and these results will be reported once these studies are complete.

Reviewer #3 (Remarks to the Author): with expertise in biostatistics, clinical trial study design

The study details three study cohorts (1A, 1B and 2A) with endpoints (DRC, ORR, CBR, and PFS). Dose evaluation for identifying DLT, AE report, PK, PD, and efficacy analysis were also described clearly.

Statistical comments for clarification:

1. Information blocked in the clinical trial protocol. There are many places blocked in the protocol which make difficult for evaluation.

Authors response: The protocol is an umbrella protocol. Module 4 (a PK fed/fasted study) and Module 1B-2 were conducted, but Module 4 will be published separately as a PK study, and Module 1B-2 will be expanded and then published as part of prostate cancer combination results. Therefore, the methodology for these modules was redacted, along with some of the optional modules that the SRC decided not to conduct. Finally, information relating to study personnel were redacted for GDPR reasons.

2. The patient numbers in the protocol (Table 1) are not matched to the manuscript (e.g., paired samples: n=9 for 240 mg + n=9 for 360mg in the protocol's Table 1 vs n=11 in the manuscript's Table 1).

Authors response: With respect to the reviewer, we believe the data may have been mis-read as the numbers n=9 for 240mg + n=9 for 360mg are incorrect. These numbers are n=5 for 240mg + n=1 for 360mg (from the 'dosed' column of Table 1 in the protocol). There are differences between the patient numbers in the protocol and the final manuscript due to the different data cut-offs for when the documents were produced. Table 1 in the protocol relates to the patient numbers recruited and dosed as of 26 November 2018 (as included in protocol version 9.0). The table in the manuscript is the most up-to-date and reflects the full final dataset. We hope this provides sufficient clarification for the differences between patient numbers provided in the protocol and final manuscript.

3. Need stratification of cancer type in Cohort 1A dose escalation. The manuscript focuses on breast cancer. However, Cohort 1A dose escalation is a mixture of breast cancer (30%) with other cancer types. Report of cohort 1A should be stratified by at least breast cancer vs non-BC for demographic, AE, and efficacy in text, table, and figure.

Authors response: We thank Reviewer 3 for this comment. Module 1A was designed to evaluate safety and tolerability during dose escalation in an advanced solid tumor patient population. It was therefore not considered necessary to focus on any particular tumor type at that stage, particularly in relation to any efficacy endpoint. Also, patient numbers would be too small to draw any conclusions regarding differences between disease types. The expansion cohorts in TNBC and HR+ breast cancer were designed to provide disease area specific safety and anti-tumor activity.

4. Figure 2 of Cohort 1B may need a boxplot with t-test or Wilcoxon test to easily visualize and test difference of treatment duration for number of prior therapy (1 vs >1). The same strategy could be also applied to Figure 3 for liver met vs non-liver met, as well as mutation (e.g., TP53)

Authors response: We thank the reviewer for this suggestion but as no analysis by the number of prior therapies was planned or conducted, we have not included this proposed statistical test. For liver metastases vs no liver metastases and TP53 mutation vs no mutation we had already applied a log-rank test – please see Figure S4 in the Supplementary Information.

5. It would be more informative to generate KM survival plot overall, liver met status, TP53 status for cohort 2A in addition to Table 3.

Authors response: We thank the reviewer for this suggestion. In fact, we already provide Kaplan Meier plots for PFS stratified by liver metastases vs no liver metastases and TP53 mutation vs no mutation in Figure S4 in the Supplementary Information.

6. Is there any association of patients experiencing only low-grade AEs with clinical outcomes or biomarker (e.g., PFS, treatment duration, or response)?

Authors response: We thank Reviewer 3 for this pertinent question. No association has been observed to date.

Reviewer #4 (Remarks to the Author): with expertise in breast cancer, CDK7

CDK7 inhibition is a promising therapeutic strategy in cancer. Samuraciclib (CT7001) as a small molecule, ATP competitive, selective oral inhibitor of CDK 7 was firstly

reported in ESMO in 2021, it has shown an acceptable safety profile with evidence of anti-tumour activity in combination with fulvestrant for patients with advanced HR+BC who have progressed on their prior CDK4/6i. The authors of this manuscript extend the evaluation of safety and anti-tumor activity of samuraciclib in the subgroup of TNBC. The authors demonstrate the special association between TP53 and samuraciclib. It would be helpful to include the further discussion of the underlying mechanism between TP53 and CDK7.

Authors response: We thank Reviewer 4 for their comments. In relation to the mechanism between TP53 and CDK7, although there are publications linking the association (already referenced in the manuscript [Discussion: Line 409]) there are no published data as to the mechanism of action. However, this will be the scope of future investigations and we have added the following text to the manuscript in relation to our own work:

Discussion: Lines 409 to 411 in the tracked changes version:

Our ongoing work indicates that CDK7 inhibition can enhance the transcriptional activity of TP53, as measured by TP53 regulated gene expression, and that this is particularly evident in HR+ breast cancer cell lines.

Moreover, TP53 is mutated in about 80% of TNBC and ESR2 is expressed in approximately in 60-80% TNBC (Key paper in this area is PMID 30990221), it may be helpful to provide the data of TP53 and CDK7 in the subgroup of TNBC.

Our preliminary analysis suggests that in the TNBC study there was no correlation between TP53 mutation status and response. However, due to the heterogeneous nature of this population and the small number of patients studied, we have not included these data in the current manuscript. We have an ongoing collaboration to study biomarkers linked to response in TNBC.

Some limitation of the analysis should not be omitted, as the nonrandomized with a relatively small number of TNBC, and there was a lack of prospective HER2 assessment description by a central laboratory additionally.

Details on the study limitations have been added to the manuscript as shown below. Central HER2 testing was not performed, but in the authors experience this is not unusual in a Phase 1 study of this type. However, all patients recruited into Module 1B-1 were defined as triple negative through local testing.

Discussion: Lines 443 to 446 in the tracked changes version: *Overall, the data show an acceptable safety profile and initial evidence of activity for samuraciclib as a selective inhibitor of CDK7. Limitations of this dataset are that the studies were uncontrolled, non-randomized, single arm evaluations in a relatively small number of patients (TNBC cohort).*

REVIEWER COMMENTS

Reviewer #1 (Remarks to the Author):

The authors have adequately addressed the comments of the reviewers.

Reviewer #3 (Remarks to the Author):

The author's response helped the reviewer better understand the study design of the manuscript. It also brings one question of the implemented 3+3 design and one recommendation of data analysis regarding TP53 mutation and liver metastasis for PFS and CBR.

1. For the classic 3+3 design, 3 patients will be enrolled in a dose cohort. If no DLT, it will escalate to a higher dose cohort with 3 patients. So, for a dose cohort without DLT, expected number of patients should be $n=3$. However, Table S5 showed $n=6-7$ in Cohort 1 and 2. Since both cohorts had no DLT, please provide explanation why there were more than 3 patients in each cohort.

2. Data analysis of TP53 mutation could confuse reader if it mixes PFS with CBR because of different denominator ($n=27$ for PFS and $n=25$ for CBR). For example, line 304-307 and S4 gives false impression that the same patients were analyzed in both PFS and CBR. Data analysis of liver metastasis has the same issue. It would be clearer if analysis of PFS and CBR are presented separately in text and also in figure. Below is the template for your consideration.

Reviewer #4 (Remarks to the Author):

The authors tried to resonse the reviewers' comments, however, it becomes a mere formality and needs to do much work thoroughly. The response to the comments are not satisfised. Much more revision should be made according to these reviewers please.

Point-by-point response to the reviewers' comments

Reviewer #3 (Remarks to the Author):

The author's response helped the reviewer better understand the study design of the manuscript. It also brings one question of the implemented 3+3 design and one recommendation of data analysis regarding TP53 mutation and liver metastasis for PFS and CBR.

1. For the classic 3+3 design, 3 patients will be enrolled in a dose cohort. If no DLT, it will escalate to a higher dose cohort with 3 patients. So, for a dose cohort without DLT, expected number of patients should be n=3. However, Table S5 showed n=6-7 in Cohort 1 and 2. Since both cohorts had no DLT, please provide explanation why there were more than 3 patients in each cohort.

Authors response: We thank reviewer 3 for their helpful comments. As written the authors recognize there is potential for a lack of clarity on the study design. The protocol did not impose a DLT restriction in the manner of a "classic" 3+3 design, but instead in this study at least 3 and up to 6 evaluable participants could be enrolled in each dose cohort in the dose escalation phase. In addition, 1 patient in Cohort 2 died due to disease progression before completing 1 full cycle of dosing, so this patient was replaced, resulting in 7 patients being recruited in this cohort.

Revised text has been added as follows (lines 108 and 109):

In this Phase I study (ClinicalTrials.gov: NCT03363893), patients with locally advanced/metastatic solid tumors received samuraciclib in dose escalation cohorts (at least 3 and up to 6 evaluable participants permitted per dose cohort~~3+3~~ design) to determine the MTD.....

And a footnote has been added to Table S5 in the Supplementary Information as follows:

^b 1 patient died due to disease progression before completing 1 full cycle of dosing, so this patient was replaced resulting in 7 patients in this cohort.

2. Data analysis of TP53 mutation could confuse reader if it mixes PFS with CBR because of different denominator (n=27 for PFS and n=25 for CBR). For example, line 304-307 and S4 gives false impression that the same patients were analyzed in both PFS and CBR. Data analysis of liver metastasis has the same issue. It would be clearer if analysis of PFS and CBR are presented separately in text and also in figure. Below is the template for your consideration.

Authors response: The authors would like to thank the reviewer for this observation and for the helpful template text provided. Based on the comments the following text has been added to the manuscript (lines 306 to 317):

There were 27 patients with TP53 mutation data available from baseline ctDNA (mutation: n=7 and no mutation: n=20). Patients with no detected TP53 mutation had longer PFS (7.4 months) compared to patients with detected TP53 mutation (1.8 months), with a hazard ratio of 0.14 for the absence of TP53 mutation (95% CI: 0.05-0.45, p-value<0.001) (Figure S4a in Supplementary

Information). Twenty-five patients with TP53 mutation data had response data (mutation: n=6 and no mutation: n=19). There were no patients with clinical benefit in the TP53 mutation group (0% CBR) and 9 patients with clinical benefit in the group with no detectable TP53 mutation (47% CBR). ~~The CBR for patients with no TP53 mutation detected in baseline ctDNA was 47% (9/19) (median PFS 7.4 months) compared with 0% (0/6) (median PFS 1.8 months) in patients with TP53 mutation (Figure S4a in Supplementary Information); hazard ratio for the absence of TP53 mutation = 0.14 (95% CI: 0.05-0.45, p-value <0.001).~~

For consistency, the description of the results for the liver metastases has also been amended in the same way, with the following text being added to the manuscript (lines 319 to 329):

For liver metastases, there were 14 patients with liver metastases at baseline and 17 patients with no detectable liver metastases at baseline. Patients with no detectable liver metastases had longer PFS (13.8 months) compared to patients with liver metastases (2.8 months), with a hazard ratio of 0.16 (95% CI: 0.04-0.59, p value<0.003) (Figure S4b in Supplementary Information). Twenty-five patients had response data (liver metastases: n=14 and no liver metastases: n=11). The CBR was 55% (6/11) for patients with no liver metastases and 21% (3/14) for those with liver metastases. ~~The CBR for patients with no liver metastases was 55% (6/11) (median PFS 11.1 months) compared with 21% (3/14) (median PFS 2.8 months) for those with liver metastases (Figure S4b in Supplementary Information); hazard ratio for the absence of liver metastases = 0.16 (95% CI: 0.05-0.59, p-value <0.003).~~

In addition, Figure S4 has been amended, with the column for CBR removed to avoid any confusion with denominators.

REVIEWERS' COMMENTS

Reviewer #3 (Remarks to the Author):

statistical issues have been addressed.